# Precision Aquaculture Drone Mapping of the Spatial Distribution of *Kappaphycus alvarezii* Biomass and Carrageenan

Nurjannah Nurdin [1,2,*], Evangelos Alevizos [3], Rajuddin Syamsuddin [4], Hasni Asis [4], Elmi Nurhaidah Zainuddin [4], Agus Aris [2], Simon Oiry [3], Guillaume Brunier [5], Teruhisa Komatsu [6,†] and Laurent Barillé [3]

1   Marine Science Department, Marine Science and Fisheries Faculty, Hasanuddin University, Makassar 90245, Indonesia
2   Research and Development Center for Marine, Coast, and Small Island, Hasanuddin University, Makassar 90245, Indonesia; agus.aris88@gmail.com
3   Institut des Substances et Organismes de la Mer, ISOMer, Nantes Université, UR 2160, F-44000 Nantes, France; evangelos.alevizos@univ-nantes.fr (E.A.); simon.oiry@univ-nantes.fr (S.O.); laurent.barille@univ-nantes.fr (L.B.)
4   Fisheries Department, Marine Science and Fisheries Faculty, Hasanuddin University, Makassar 90245, Indonesia; rajuddin.syamsuddin09@gmail.com (R.S.); hasni_yulianti@yahoo.com (H.A.); elmi18id@yahoo.com (E.N.Z.)
5   BRGM French Geological Survey, Cayenne 97300, French Guiana; guillaume.brunier@univ-nantes.fr
6   Atmosphere and Ocean Research Center Institute, The University of Tokyo, Kashiwa 277-8564, Japan; cymodocea@gmail.com
*   Correspondence: nurjannahnurdin@unhas.ac.id; Tel.: +62-8138-739-0203
†   Currect Address: Japan Fisheries Resource Conservation Association, Tokyo 104-0044, Japan.

**Abstract:** The aquaculture of *Kappaphycus alvarezii* (*Kappaphycus* hereafter) seaweed has rapidly expanded among coastal communities in Indonesia due to its relatively simple farming process, low capital costs and short production cycles. This species is mainly cultivated for its carrageenan content used as a gelling agent in the food industry. To further assist producers in improving cultivation management and providing quantitative information about the yield, a novel approach involving remote sensing techniques was tested. In this study, multispectral images obtained from a drone (Unoccupied Aerial Vehicle, UAV) were processed to estimate the fresh and carrageenan weights of *Kappaphycus* at a cultivation site in South Sulawesi. The UAV imagery was geometrically and radiometrically corrected, and the resulting orthomosaics were used for detecting and classifying *Kappaphycus* using a random forest algorithm. The classification results were combined with in situ measurements of *Kappaphycus* fresh weight and carrageenan content using empirical relations between the area and weight of fresh seaweed/carrageenan. This approach allowed quantifying seaweed biometry and biochemistry at single cultivation lines and cultivation plot scales. Fresh seaweed and carrageenan weights were estimated for different dates within three distinct cultivation cycles, and the daily growth rate for each cycle was derived. Data were upscaled to a small family-scale farm and a large-scale leader farm and compared with previous estimations. To our knowledge, this study provides, for the first time, an estimation of yield at the scale of cultivation lines by exploiting the very high spatial resolution of drone data. Overall, the use of UAV remote sensing proved to be a promising approach for seaweed monitoring, opening the way to precision aquaculture of *Kappaphycus*.

**Keywords:** aquaculture; carrageenan; drone; *Kappaphycus*; multispectral; seaweed; UAV

## 1. Introduction

Indonesia is the world's second-largest producer of seaweed after China, and the leading producer of red seaweed, notably the group of eucheumatoids, which refer to the two

genera *Kappaphycus* and *Eucheuma* [1]. Eucheumatoids are commercially farmed seaweed harvested for various applications, including food products, cosmetics and pharmaceuticals. Seaweed farming is one of the priorities for development in the country due to the increasing global demand for raw and processed seaweed, especially red seaweed [2,3]. Indonesia has considerable potential to increase marine aquaculture [4], estimated at 12 million ha [5]. The southern part of Sulawesi Island is the largest of Indonesia's red seaweed-producing regions, where artisanal farming is mainly practiced in the close vicinity of the coastline in shallow areas [6]. *Kappaphycus alvarezii* ((Doty) Doty ex P.C.Silva, 1996) is the main species cultivated in this region (*Kappaphycus* hereafter). Its widespread aquaculture in coastal communities is attributable to the ease of cultivation, harvesting and drying techniques, and the short production cycle of approximately 35–45 days [2]. It is a red seaweed from the class of Florideophyceae, but has many phenotypes with various shapes and colors, from red to green [7,8]. This species is mainly cultivated for its carrageenan content, which is used as a gelling agent for food processing in the industry [9,10].

In South Sulawesi, the most popular method of cultivating *Kappaphycus* is the simple and cost-effective long line technique [11]. It involves suspending a series of lines (i.e., ropes) in the water column, using plastic bottles as floats, and attaching small propagules of *Kappaphycus* to the lines, tied at ca. 20 cm intervals along the lines. The propagule grows on the lines and is harvested after ca. 40 days of cultivation [12]. These propagules are obtained by taking small fragments of existing thalli that will grow by vegetative propagation. Seaweed cultivation plots generally occupy large areas (at least several 100 s m$^2$) and production can be influenced by a number of factors. Most importantly, the aquaculture of *Kappaphycus* is susceptible to diseases such as ice-ice disease, which can cause significant crop damage and economic losses for farmers [13]. In addition, seasonality of environmental stressors, such as temperature, salinity and water quality, can negatively impact the growth and survival of this species, resulting in reduced yields [14–16]. Careful monitoring of seaweed cultivation parameters (i.e., biomass, growth rate, water parameters) means that cultivation management can be adapted, therefore it is required to better control biomass production and carrageenan content [2,7]. Many studies that measured the growth and production of *Kappaphycus* were based on sampling a limited number of individual thalli (e.g., [14]), while more rarely measurements have been taken from long lines [7]. Generally, the main challenges in seaweed aquaculture monitoring include: (a) laborious and time-consuming in situ measurements [14], (b) large-scale, remote, heterogeneous cultivation areas that need to be surveyed [17,18], and (c) mixed sampling techniques that often produce inconsistent results among different areas [7]. Consequently, in order to tackle these challenges, a novel remote sensing application is investigated in the present study.

Remote sensing techniques—including satellite and aerial imaging—have been applied to various aquatic (either marine or freshwater) vegetation and benthic cover mapping studies during the last few decades [19–21]. Nevertheless, an increasing number of emerging, remote sensing applications rely on drone technology for identifying aquatic vegetation types. Notably, several recent studies have utilized drone-based multispectral imagery for mapping seaweed and macroalgae [22,23], kelps [24–26] and intertidal reefs [27,28]. Drones with multispectral sensors provide an affordable tool for collecting centimeter-resolution imagery at frequent/on-demand time intervals and preferentially when environmental conditions are optimal. In addition, these multispectral sensors record narrow-band images in the visible and infrared ranges corresponding to the wavelengths of vegetation spectra, allowing for identifying floating seaweed [25,29], and thus, being a promising tool in seaweed aquaculture mapping. Additionally, drone multispectral sensors share similar wavelength bands with satellite sensors, such as the Sentinel-2, allowing for inter-comparisons and data upscaling [30,31]. However, drone imagery tends to yield a relatively low signal-to-noise ratio due to the high resolution of imaging geometry. When collecting data above water surfaces, the most common noise sources include sun glint and water turbidity. Thus, image acquisition should occur under favorable optical conditions, or image pre-processing and filtering should be applied before any analysis. Modern

technological advances in drone and sensor equipment have resulted in the development of a new sector, that of 'precision aquaculture', and a growing demand for automated procedures in aquaculture production [32–34]. For example, [32] estimated the canopy area of offshore-farmed kelp species using drone imagery, while [35] applied drone imagery to monitor green macroalgae cultivation in the Yellow Sea. The application of drones in aquaculture appears to have strong potential due to their flexibility in data acquisition and the high resolution of resulting imagery. To our knowledge, there has been no synoptic mapping yet of *Kappaphycus* aquaculture at a relevant spatial resolution that would allow the detection of seaweed attached to the cultivation lines. This is in striking contrast to agriculture, where drones have been increasingly used to monitor crops, gather data and make better decisions about managing the fields [36,37].

This study aims to apply drone technology for high-precision monitoring of fresh weight and carrageenan weight of *Kappaphycus* seaweed during three cultivation cycles in South Sulawesi (Indonesia). A multispectral sensor was used for acquiring very high spatial resolution images of seaweed attached to long lines at regular intervals during each cycle. Additional in situ measurements of individual seaweed fresh weight, biometry and carrageenan content were used to obtain scaled-up relationships to drone images. Abiotic variables were also collected for each date within a cultivation cycle to describe the environmental conditions. An extensive set of multispectral drone images was processed, combining geospatial and machine learning techniques, to propose a workflow using a random forest algorithm for image classification and subsequent geospatial analyses. This workflow was used to produce spatial distribution maps of *Kappaphycus* fresh weight and carrageenan weight and to analyze their changes within a cultivation plot during three, forty-day cultivation cycles.

## 2. Materials and Methods

The study was located in Punaga village (5°35′2.257″S, 119°25′52.058″E), Mangarombang District, Takalar Regency, South Sulawesi, Indonesia (Figure 1C). Field data collections were conducted from March to September 2022 and consisted of (1) seaweed biometric and biochemical measurements (fresh weight, length, width and carrageenan content), (2) UAV data collection and (3) water quality measurements. The cultivation cycles were ca. 40 days but farmers could harvest their crops earlier. Two cycles covered the transition period from the rainy to the dry season (cycle 1: March–April and cycle 2: April–May). Another cycle was followed during the dry season (cycle 3: June–July). For each cycle, measurements were planned at five time intervals every ten days: $t_0$, $t_{10}$, $t_{20}$, $t_{30}$ and $t_{40}$. This 10-day interval was sometimes shorter or longer depending on field accessibility and farmer practices.

### 2.1. Cultivation, Biometry and Carrageenan Content

The farming system used in South Sulawesi is the long line method (Figure 1D). At the farming site, a typical cultivation plot consists of several nylon lines of 20 m or 25 m in length and 0.5 m apart. These contain individual thalli of *Kappaphycus* with an initial weight between 20–30 g each at the start of the cultivation cycle. The thalli were attached at ca. 20–25 cm intervals, giving a maximum of 125 seaweed thalli for a 25 m long line. Plastic bottles used as floats were attached at every 5 m of the line (Figure 1D), and anchors were used to secure the infrastructure at the extremities of the line. For each cycle, five lines were randomly chosen inside a cultivation plot and identified with red buoys. For each line, three replicates of *Kappaphycus* were identified and repeatedly measured for each time interval of ca. 10 days over the three cycles. Fifteen individual samples were monitored per date (Figure 1B). The length and width were used to assess the seaweed area that would be observed from the drone sensor. The drone sensor can capture the seaweed even from the first cultivation day when the sea state is relatively calm and water clarity is good. The fresh weight was obtained with a field scale after leaving the sample draining for ten minutes on soft tissue. The dry weight was obtained in the laboratory after drying the

samples for 3 days at 60 °C in an oven. Fresh weight was used to calculate the individual Specific Growth Rate (SGR) expressed as the daily growth rate (%·day$^{-1}$) [38]; 216 samples were used to obtain a relationship between seaweed surface (cm$^2$) and fresh weight (g). For each cycle and time interval within a cycle, a minimum of five additional samples were randomly collected for carrageenan content and characterized for their biometry (fresh weight, surface area). Carrageenan was measured according to the Indonesian National Standard (SNI 01-4498-1998) and expressed as the percentage of the dry weight of samples. Fresh seaweed was cleaned and washed with freshwater, cut into small pieces and then soaked in water for 24 h to remove salt and sediment. It was then dried as previously indicated. The analysis used sodium hydroxide extraction (NaOH) followed by precipitation with ethanol. Precipitated carrageenan were dried at 70 °C for 4 h. Dried carrageenan powder was weighed and compared to the dry weight of the seaweed. More details can be found in [39]. Carrageenan content (%) was calculated as:

$$\text{Carrageenan content (\%) = weight of carrageenan (g)} \times 100/\text{dry weight of sample (g)}$$

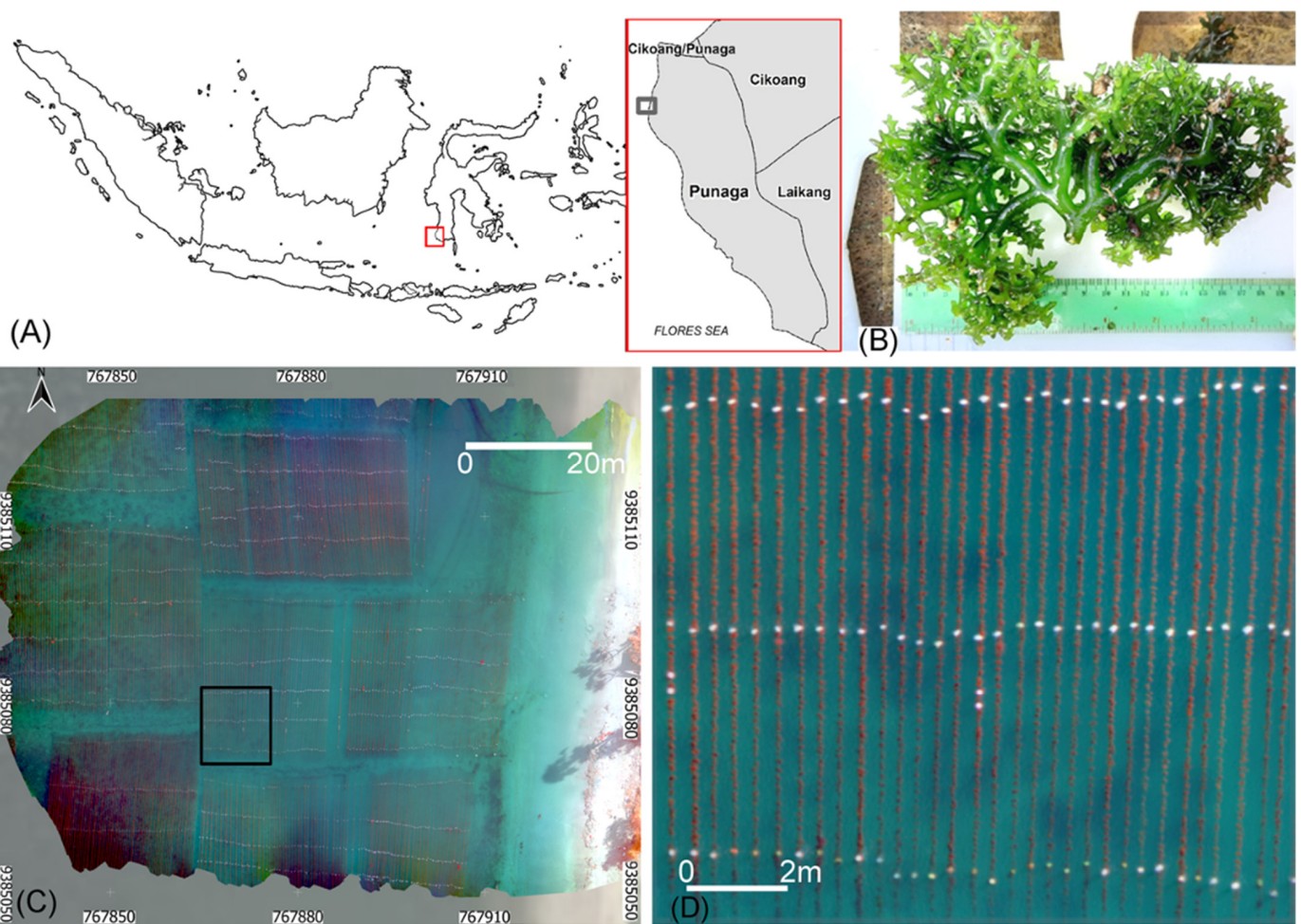

**Figure 1.** (**A**) Indonesian archipelago with the red rectangle indicating South West Sulawesi; the sub-district of Punaga with the black rectangle showing the study area. (**B**) Individual thalli of the green variant of *Kappaphycus alvarezii* cultivated in Punaga. (**C**) False color orthomosaic of the farming area showing cultivation plots (=parcels) with a variable number of long lines. (**D**) Close-up corresponding to the black rectangle in C; ca. 32 lines with *Kappaphycus* can be seen. Plastic bottles used as floats appear white.

### 2.2. UAV Data Acquisition

UAV images were obtained using a DJI Phantom 4 with a Real-Time Kinematic Differential GPS (RTK D-GPS). During each cultivation cycle, images were acquired at a 10-day interval from the beginning of the cycle ($t_0$) to the last day of harvest ($t_{40}$) along with in situ measurements of *Kappaphycus*. The DJI Phantom 4 multispectral camera is a global shutter sensor of six separate lenses and CMOS comprising five spectral bands and one RGB sensor at 2.12 MP resolution. It is equipped with a 5.27 mm fixed focal lens (equivalent in 35 mm format to a 40 mm focal lens), mounted on gimbals stabilized on three axes. The multispectral bands are monochrome sensors with a spectral range including blue ($450 \pm 16$ nm), green ($560 \pm 16$ nm), red ($650 \pm 16$ nm), red edge ($730 \pm 16$ nm) and near-infrared ($840 \pm 26$ nm) wavelengths. The camera system includes a sun incident light sensor on the top of the UAV that is combined with the UAV's internal GPS receivers. The flight plans were designed using the DJI Ground Station Professional application and by taking into account the specifications of the multispectral camera. The flight height was 30 m with a Ground Size Dimension (GSD) image pixel of 1.6 cm. The camera orientation was set to obtain nadir photographs with a frontal overlap of 90% and a lateral overlap of 80%. To minimize the sun glint contamination, the flight paths were oriented perpendicular to the sun early in the morning. The shooting interval was set to two seconds. The radiometric calibration of the camera was computed in post-processing from UAV EXIF file settings written into image metadata and sunlight sensor records.

### 2.3. UAV Image Processing

Aerial imagery was processed with proprietary software Pix4D 4.5$^{©}$ and was further analyzed using SAGA 8.0 GIS software [40]. The overall workflow is presented in the diagram in Figure 2. Initially, the positions of cameras were reconstructed using structure-from-motion (SFM) processing and camera-specific geometric corrections were applied. Following this, bundle adjustment and georeferencing of orthomosaics were performed using the navigation metadata of imagery. The final orthomosaics from each band were radiometrically corrected using the EXIF metadata regarding sun angle and incoming solar radiance. After processing, the orthomosaics were imported to SAGA GIS for image analysis. An additional raster was created by calculating the ratio of the Blue over the Red-Edge band. Not all temporal datasets were suitable for processing; thus, only selected days from each cultivation cycle were processed and further analyzed.

### 2.4. Image Classification

#### 2.4.1. Algorithm Introduction

The random forest (RF) classification algorithm [41] was applied for discriminating *Kappaphycus* from background bottom and plastic flotation objects. The concept of the RF algorithm is based on an ensemble procedure of multiple random subsets (classification trees) of the explanatory variables (predictors) for generating a classification model describing the variability of the dependent variable (classes). Thus, a set of training data is mandatory in the RF process, and it should capture as much of the data variability for each class in the study area. During the model building (i.e., training), the RF reserves randomly selected parts of the training data for internal cross-validation of the results (out-of-bag sample). One explanatory variable is neglected at each iteration, and its importance score is calculated by assessing the prediction error. The variable importance calculation assists further with interpretations about which predictors show the most significant influence in identifying each class. The RF was preferred for its high accuracy, insensitivity to overfitting and availability in many standard software [42]. It has been successfully applied in several studies in marine habitat mapping and seafloor characterization studies [43–47].

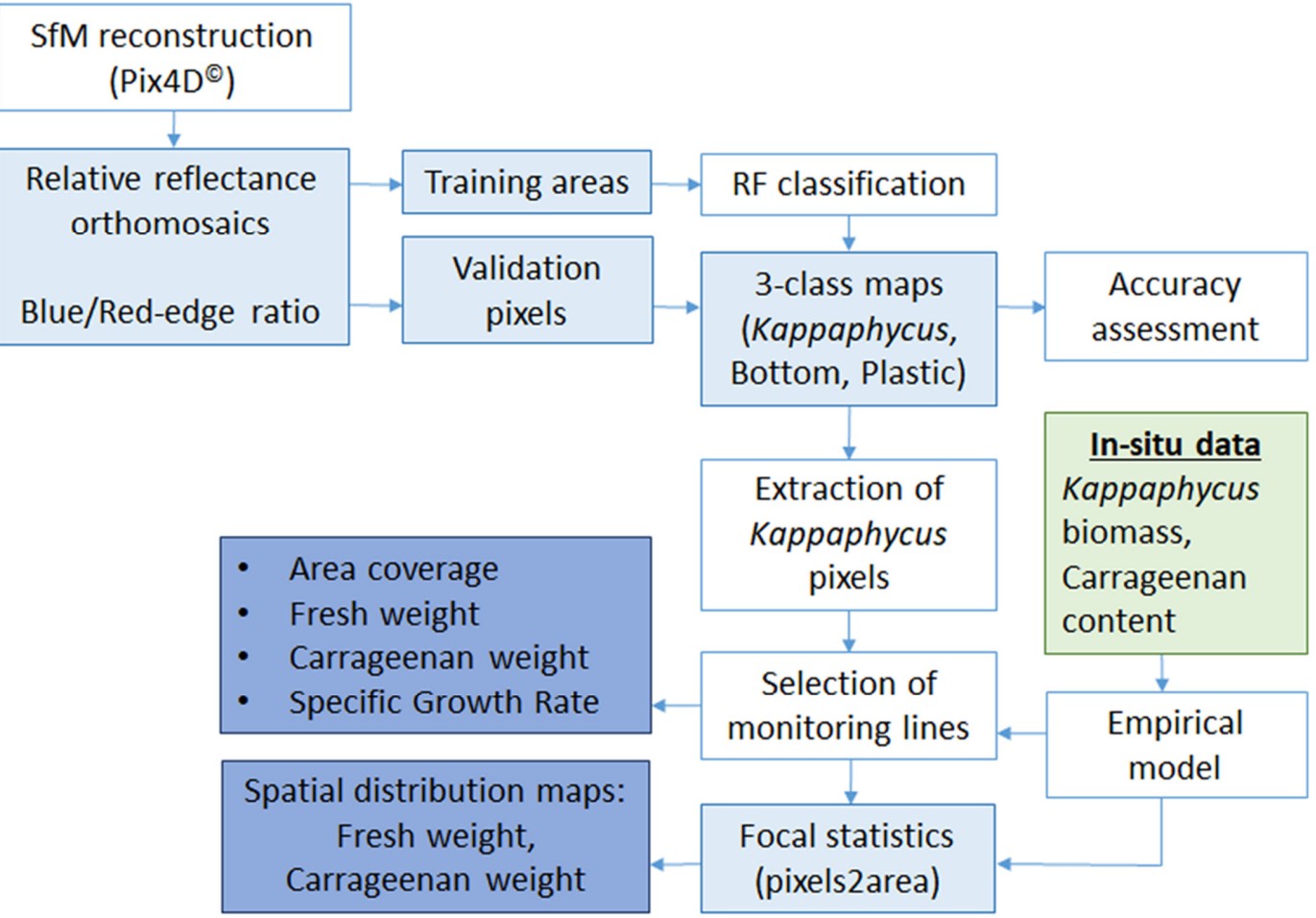

**Figure 2.** Workflow diagram including image processing, classification and geospatial analyses tasks followed in this study.

### 2.4.2. Training/Validation Data

Initially, one training set was created for each temporal dataset of each cultivation cycle. Training polygons (Figure 3A) were digitized in SAGA GIS based on the visual interpretation of RGB composites (compiled from individual orthomosaics). Apart from the five spectral bands of the drone camera, the band ratio between the Blue and the Red-Edge bands was calculated. These bands were selected since they show a strong reflectance contrast in the *Kappaphycus* in situ spectrum and the blue reflectance dominates the bottom areas in general. Consequently, this band ratio was applied for enhancing the differentiation between the floating *Kappaphycus* from background bottom types. A similar band ratio (Blue/Red) has also been proposed for floating kelp canopy detection by [25]. Data exploration was performed to identify how well each class (bottom, *Kappaphycus* and plastic) was separated by the six predictor variables (See Appendix A Figure A1 for a typical example of the class separation between the three classes and the relative reflectance orthomosaics).

### 2.4.3. Random Forest Implementation

The ViGrA library was used to implement the RF in SAGA GIS [48] using: (a) 1000 trees for training the model, (b) subsampling with replacement and (c) using a number of variables per node split equal with the square root of the total variables. The classification maps were validated using a separate set of single pixels resulting from a visual examination of the RGB composites. A few hundred pixels were extracted from the boundaries of the classified objects by applying a quasi-regular type of sampling (Figure 3B). Selecting

polygons with several pixels assists in having a robust training set; however, selecting polygons for validation would lead to an underestimation of misclassifications in this study. Therefore, by selecting single pixels, the objectivity of the validation procedure was increased and spurious results could be effectively captured. The ratio between training/validation pixels is approximately 75/25.

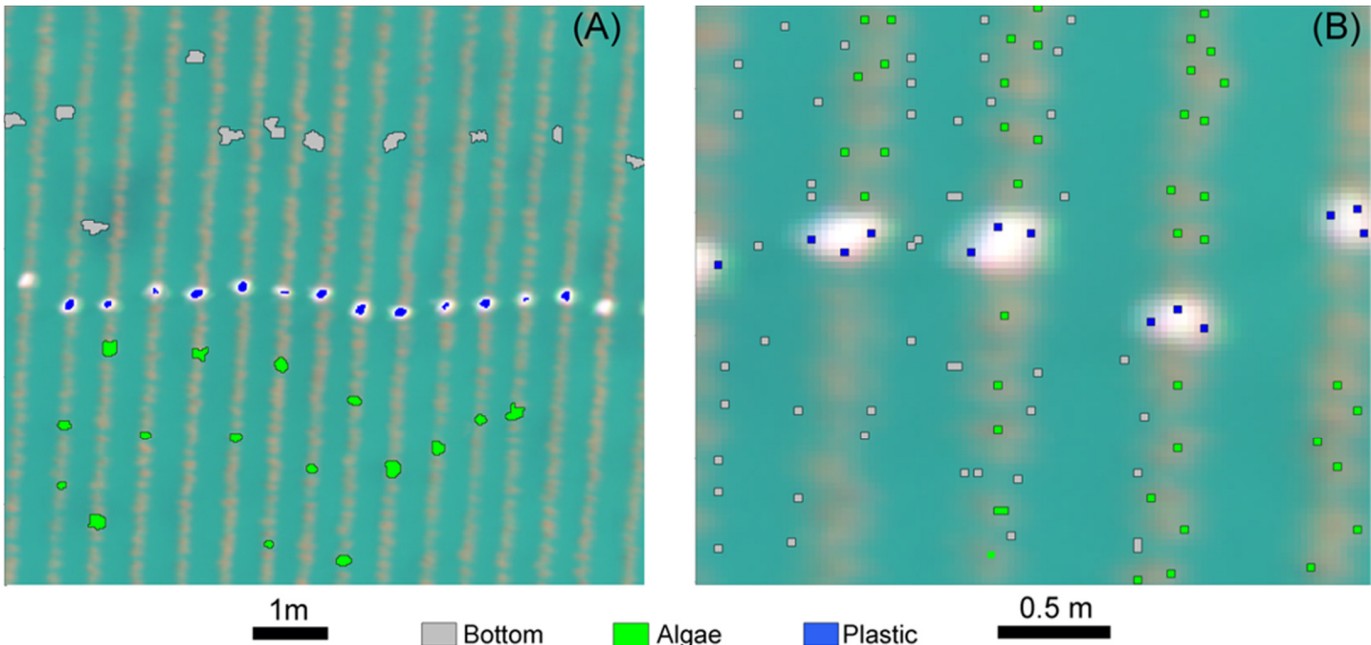

**Figure 3.** False color close-ups of cultivation lines overlaid with an example of: (**A**) Training polygons, (**B**) Validation set with pixel-based interpretations.

### 2.4.4. Classification Accuracy Assessment

Validation of RF class predictions was assessed by calculating the confusion matrix and the kappa coefficient (Equation (1)) of the agreement for each temporal dataset.

$$K = P_o - P_e/1 - P_e \tag{1}$$

where, $P_o$ is the probability of agreement and $P_e$ is the probability of random agreement.

After classification, the *Kappaphycus* class was extracted from each temporal dataset for further quantitative mapping of fresh weight and carrageenan weight. Although the orthomosaics captured a large area, cultivation lines can be modified/relocated by farmers during a cultivation cycle. Thus, a subset of lines was preserved for monitoring purposes and identified with buoys within a selected cultivation plot (Figure 1D).

### 2.5. *Geospatial Analyses*

An assessment of *Kappaphycus* fresh weight per line was performed by calculating the number of *Kappaphycus* class pixels on a neighborhood corresponding to the width of each cultivation line. This technique allows for visualizing the natural, along-line variations of seaweed growth and uses a sufficiently small analysis window so that only pixels from each line are used for producing the spatial distribution maps. The sum of those pixels was then converted to the area of the *Kappaphycus* surface, which was subsequently converted to fresh weight using the empirical relationship obtained from in situ data (Equation (2)). Fresh weight was converted to carrageenan weight, using the carrageenan content and the fresh weight/dry weight ratio of in situ samples collected during each cycle. Using *Kappaphycus* fresh weight per line at the beginning and the end of a cultivation cycle, the SGR (%) was calculated and compared with SGR derived from in situ samples. Production per line was similarly estimated as the difference in fresh weight between $t_0$ and $t_{40}$ and

expressed in $g \cdot m^{-1}$ per cycle [7]. *Kappaphycus* fresh weight and carrageenan weight were expressed in $g \cdot m^{-2}$ to display their spatial distribution along lines but were converted per linear meter of line $(g \cdot m^{-1})$, a common unit in the literature [7]. Between four and eight monitoring lines were analyzed to estimate growth and biomass metrics.

## 2.6. Statistical Analysis

After checking normality with the Shapiro-Wilk test, the comparison of SGR and carrageenan content between the three cycles was tested with a Kruskal-Wallis non-parametric test. All univariate tests were performed using PAST 3.25 [49] and R [50]. A power model was adjusted between fresh weight and seaweed area. The goodness of fit of this empirical relationship was assessed with the coefficient of determination $r^2$, the Root Mean Square Error (RMSE) and the Mean Absolute Error (MAE).

## 3. Results

### 3.1. Biometry and Carrageenan Content

For each of the three cycles, the fresh weight of individual *Kappaphycus* thalli had increased by approximately ten at the end of the forty days of cultivation. From initial mean weights ranging from 24.0 to 30.7 g, final average weights from 268 g to ca. 300 g were obtained (Table 1). The SGR significantly decreased from $6.9\% \cdot d^{-1}$ for cycle 1 to $5.4\% \cdot d^{-1}$ for cycle 3 (Kruskal-Wallis, $p < 0.01$, Table 1). The average carrageenan content increased within cultivation cycles 1 and 3 from the start ($t_0$) to the last date ($t_{40}$) (Table 1). The carrageenan content (quoted values are standard deviation) at harvest ($t_{40}$) significantly increased gradually from cycle 1 with $54.9 \pm 2.8\%$, to cycle 2 with $55.7 \pm 4.5\%$ and cycle 3 with $61.6 \pm 2.3\%$ (Kruskal-Wallis, $p < 0.01$, Table 1). These in situ measurements were used to calculate an empirical relationship (Equation (2)) relating fresh weight (FW) to seaweed area (A) (Figure 4):

$$FW = 0.014 \times A^{1.65} \tag{2}$$

This power model, characterized by an $r^2 = 0.81$, an RMSE = 44 g and an MAE = 34 g, was subsequently used to convert *Kappaphycus* pixels area to fresh weight.

**Table 1.** Dates of the three cultivation cycles of *Kappaphycus* monitored in 2022 in Punaga (South Sulawesi) and corresponding in situ measurements. FW = Fresh Weight (average of individual thalli $n = 15$), SGR = Specific Growth Rate ($n = 15$), Carrageenan ($n = 5$). Mean $\pm$ standard deviation.

| Cycle | Date | FW | SGR | Carrageenan |
|---|---|---|---|---|
| | | (g) | (%) | (%) |
| 1 | 10/03 | 24.0 ± 2.1 | | 48.8 ± 4.3 |
| 1 | 29/03 | 103.0 ± 24.7 | | 48.4 ± 2.4 |
| 1 | 10/04 | 229.3 ± 50.9 | | 57.3 ± 4.8 |
| 1 | 16/04 | 299.3 ± 51.4 | 6.9 ± 0.7 | 54.9 ± 2.8 |
| 2 | 19/04 | 24.0 ± 2.0 | | 61.2 ± 1.5 |
| 2 | 28/04 | 50.7 ± 11.6 | | 56.6 ± 4.2 |
| 2 | 10/05 | 104.7 ± 23.5 | | 54.1 ± 4.9 |
| 2 | 20/05 | 180.7 ± 41.1 | | - |
| 2 | 30/05 | 268.0 ± 60.7 | 5.8 ± 0.7 | 55.7 ± 4.5 |
| 3 | 08/06 | 30.7 ± 6.2 | | 53.6 ± 1.2 |
| 3 | 17/06 | 68.3 ± 11.9 | | 55.4 ± 4.5 |
| 3 | 27/06 | 120.0 ± 37.6 | | 55.6 ± 2.6 |
| 3 | 08/07 | 201.3 ± 68.5 | | 60.8 ± 2.7 |
| 3 | 19/07 | 301.4 ± 77.0 | 5.4 ± 0.8 | 61.6 ± 2.3 |

### 3.2. Random Forest Classification

Predictive mapping of three major classes (*Kappaphycus*, bottom and plastic buoys) of drone imagery was separately performed for each date and cultivation cycle. Accuracy assessment was applied for each prediction map, resulting in an equal number of confusion

matrices. The confusion matrices for each cycle were summarized and presented in Table 2. Most prediction accuracies were more than 87%, with kappa agreement values greater than 0.77. Particularly, in cycle 1 and cycle 3 classifications, *Kappaphycus* achieved high user accuracy (~88%), while in cycle 2, it was mapped with the highest user accuracy of 92%. The most common misclassification of *Kappaphycus* was with plastic bottles. This is probably because seaweed and semi-submerged plastic buoys similarly reflect in Red-Edge and NIR wavelengths. The random forest algorithm reported the importance scores of each predictor variable (i.e., orthomosaics) on classification performance. The ranking of each variable (based on the Gini decrease score) from all classification runs was summarized, and the average ranking of each predictor was calculated. The overall ranking of predictors suggests that the blue/red-edge ratio was the most important predictor, followed by the Red-Edge band (Figure 5). These results confirm the data exploration outputs (Figure A1, Appendix A) where the three classes appeared to be optimally separated by the Blue/Red-edge ratio and the Red-Edge orthomosaics.

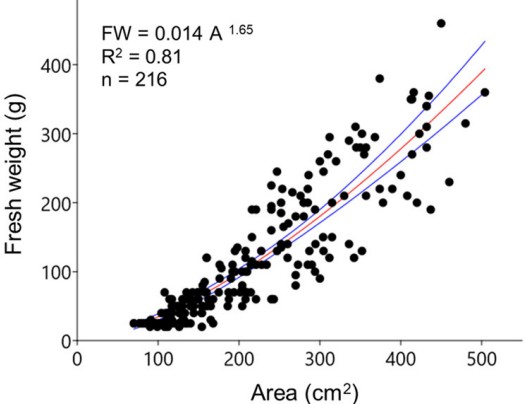

**Figure 4.** Relationship between *Kappaphycus* area (cm$^2$) and fresh weight (g) obtained from in situ samples (dots) collected during three cultivation cycles in Punaga (South Sulawesi). The power model (red line) is represented with its 95% confidence intervals (blue lines).

*3.3. Spatial Distribution of Fresh Weight and Carrageenan*

Classification of drone imagery combined with the empirical relationships allowed for capturing the spatial distributions of *Kappaphycus* fresh weight and carrageenan weight at the level of individual lines within a cultivation plot (Figure 6). From the false-color composite mosaic of a cultivation plot (Figure 6A), the RF classification discerned three classes: *Kappaphycus*, plastic bottles and bottom (Figure 6B). The *Kappaphycus* class was then converted to fresh weight (Figure 6C) and weight of carrageenan (Figure 6D). The within-line variability of the spatial distribution of fresh weight and carrageenan showed that the largest values were observed towards the center of the lines (Figure 6C,D). In addition, it was possible to estimate the fresh weight and carrageenan weight per meter of the cultivation line (g·m$^{-1}$; quoted values are standard deviations) for comparison with the literature data (Table 3 and Figure A2, Appendix A). By subsampling several lines within the cultivation plot that were retained for monitoring, it was possible to estimate the growth rate of seaweed at the scale of a line. For cycle 1, cultivation started with an average fresh weight of $550 \pm 128$ g·m$^{-1}$ at $t_0$, leading to a final average fresh weight of $1912 \pm 356$ g·m$^{-1}$ at $t_{40}$ (Table 3). For cycle 2, the initial ($t_0$) fresh weight was $344 \pm 157$ g·m$^{-1}$ on average, showing a rapid growth until $t_{10}$ with an average fresh weight of $1019 \pm 343$ g·m$^{-1}$ to reach a final yield of $2151 \pm 227$ g·m$^{-1}$ at $t_{40}$. Cycle 3 was characterized by an initial fresh weight of $456 \pm 141$ g·m$^{-1}$ but reached a lower yield at $t_{40}$ of $1435 \pm 388$ g·m$^{-1}$. There was an increase in carrageenan weight between $t_0$ and $t_{40}$ for four individual lines within a cultivation plot (Figure 7). The total weight of carrageenan per meter of cultivation line varied between 570 and 970 g at the start of the cultivation, and the final weight at the end of the cultivation cycle was from 2720 to 3580 g (Figure 7). The carrageenan per cultivation

line increased from 27 to 105 g·m$^{-1}$ for cycle 1, from 21 to 120 g·m$^{-1}$ for cycle 2 and from 24 to 88 g·m$^{-1}$ for cycle 3. The SGR estimated with drone data from individual lines was lower than the SGR estimated from individual *Kappaphycus* samples ranging from 3.6%·d$^{-1}$ for cycle 3 to 4.9%·d$^{-1}$ for cycle 2 (Table 3). However, cycle 3 SGR was lower than cycles 1 and 2, as observed with individual thalli. The fresh weight net production ranged from 903 $\pm$ 314 g·m$^{-1}$ to 1807 $\pm$ 296 g·m$^{-1}$.

**Table 2.** Random Forest classification confusion matrices for each cultivation cycle. Each matrix is composed by adding and averaging the results of individual matrices corresponding to temporal drone datasets. Numbers in bold are true positives.

| **Cycle-1** | | | | | |
|---|---|---|---|---|---|
| CLASS | Kappaphycus | Bottom | Plastic | Sum User | User accuracy% (mean) |
| Kappaphycus | **170** | 0 | 17 | 187 | 88.5 |
| Bottom | 11 | **281** | 6 | 298 | 94.5 |
| Plastic | 0 | 0 | **76** | 76 | 100.0 |
| Sum Producer | 181 | 281 | 99 | | |
| Producer accuracy% (mean) | 90.5 | 100.0 | 76.7 | | |
| **Cycle-2** | | | | | |
| CLASS | Kappaphycus | Bottom | Plastic | Sum User | User accuracy% (mean) |
| Kappaphycus | **482** | 5 | 32 | 519 | 92.0 |
| Bottom | 41 | **442** | 12 | 495 | 90.6 |
| Plastic | 0 | 0 | **139** | 139 | 100.0 |
| Sum Producer | 523 | 447 | 183 | | |
| Producer accuracy% (mean) | 91.0 | 97.6 | 74.8 | | |
| **Cycle-3** | | | | | |
| CLASS | Kappaphycus | Bottom | Plastic | Sum User | User accuracy% (mean) |
| Kappaphycus | **238** | 7 | 25 | 270 | 87.9 |
| Bottom | 19 | **112** | 25 | 156 | 78.8 |
| Plastic | 0 | 0 | **70** | 70 | 100.0 |
| Sum Producer | 257 | 119 | 120 | | |
| Producer accuracy% (mean) | 93.8 | 94.8 | 59.1 | | |
| **OVERALL** | | | | | |
| | | Cycle-1 | Cycle-2 | Cycle-3 | |
| Accuracy% (mean) | | 94.0 | 91.1 | 85.9 | |
| Kappa (mean) | | 0.9 | 0.85 | 0.77 | |

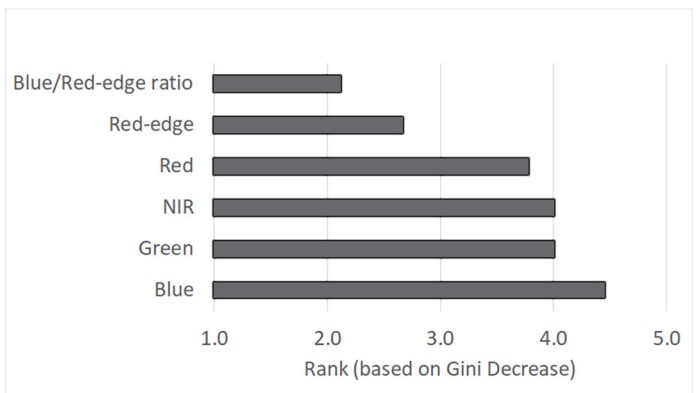

**Figure 5.** Average ranking of predictor variable importance, based on the Gini decrease score resulting from nine classification runs (one for each date with drone data, Table 3). 1: high importance, 5: low importance.

**Table 3.** Drone-derived biomass metrics with corresponding dates of in situ measurements for three cultivation cycles in Punaga (South Sulawesi) in 2022. Averaged values were calculated from individual lines (*n* varied from four to eight lines). FW = Fresh Weight; BNP = Biomass Net Production; SGR = Specific Growth Rate. Mean ± standard deviation.

| Cycle | Date | FW Drone | Carrageenan Drone | BNP Drone | SGR Drone |
|:---:|:---:|:---:|:---:|:---:|:---:|
| | | $(g \cdot m^{-1})$ | $(g \cdot m^{-1})$ | $(g \cdot m^{-1})$ | (%) |
| 1 | 15/03 | 550 ± 128 | 27 ± 6 | | |
| 1 | 29/03 | - | - | | |
| 1 | 10/04 | - | - | | |
| 1 | 16/04 | 1912 ± 356 | 105 ± 19 | 1516 ± 225 | 4.7 ± 0.4 |
| 2 | 19/04 | 344 ± 157 | 21 ± 10 | | |
| 2 | 28/04 | 1019 ± 343 | 58 ± 19 | | |
| 2 | 10/05 | - | - | | |
| 2 | 20/05 | 1945 ± 266 | 97 ± 15 | | |
| 2 | 30/05 | 2151 ± 227 | 120 ± 13 | 1807 ± 296 | 4.9 ± 1.0 |
| 3 | 08/06 | 456 ± 141 | 24 ± 8 | | |
| 3 | 17/06 | 810 ± 159 | 45 ± 9 | | |
| 3 | 27/06 | - | - | | |
| 3 | 08/07 | - | - | | |
| 3 | 19/07 | 1435 ± 388 | 88 ± 24 | 903 ± 314 | 3.6 ± 0.5 |

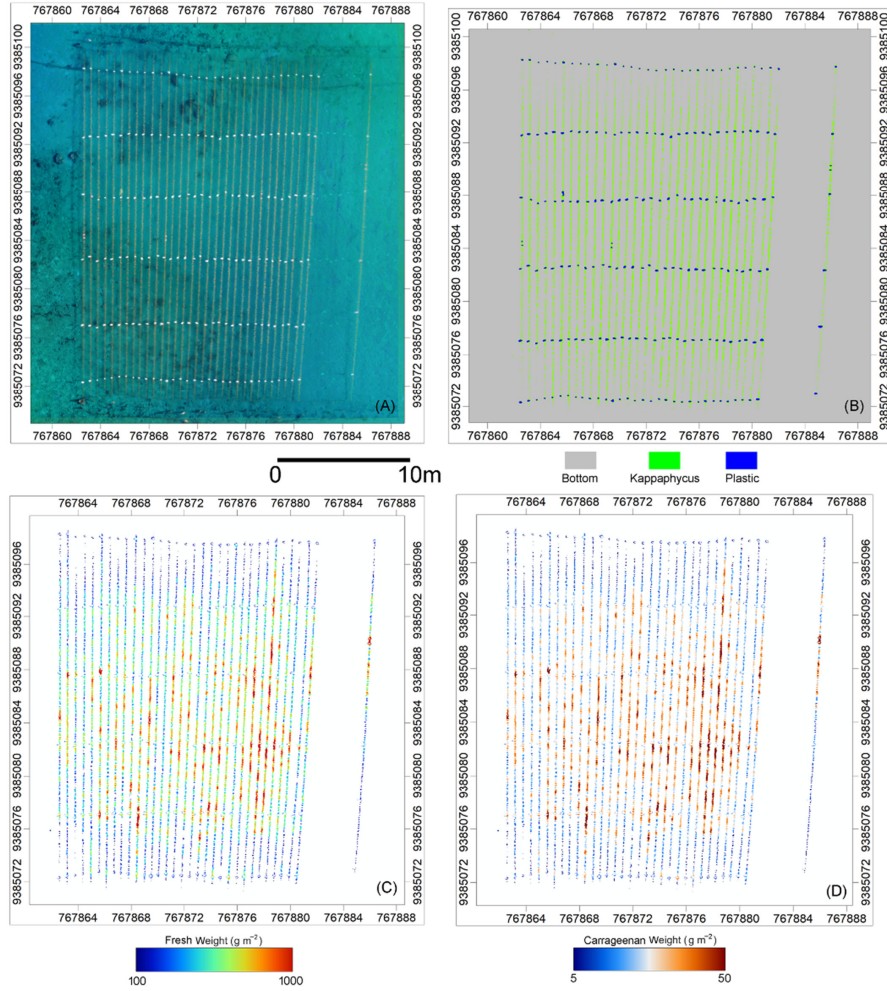

**Figure 6.** Mapping of a *Kappaphycus* cultivation plot. The cultivation plot has 32 lines of 25 m; an isolated line can be seen on the right part of each image. (**A**) False-color mosaic of the first date ($t_0$) of cycle 1. (**B**) Random forest classification of the scene, (**C**) Spatial distribution of fresh weight per unit area, (**D**) Spatial distribution of carrageenan weight per unit area. The area is defined by a neighborhood of a 20 cm radius around each pixel.

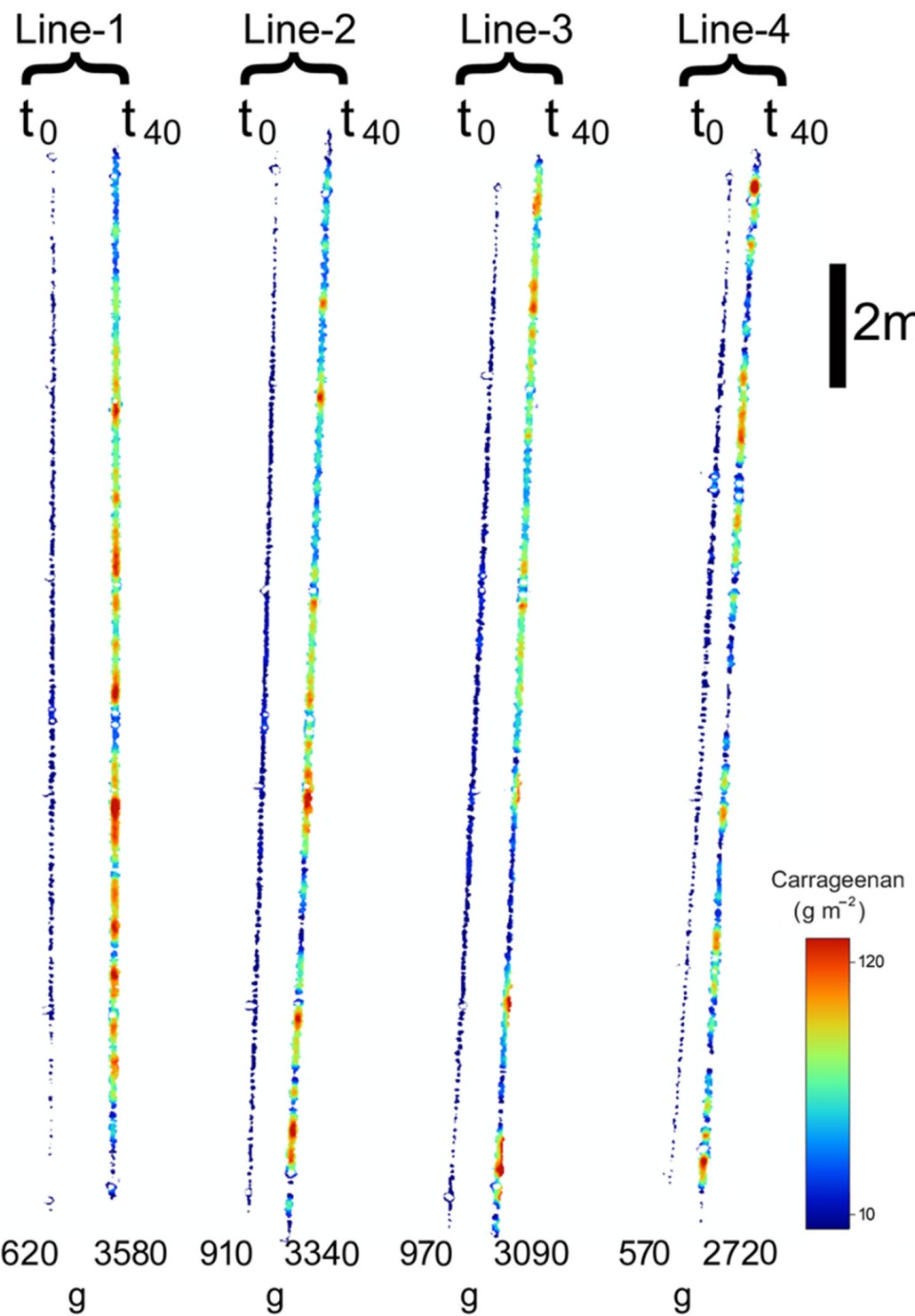

**Figure 7.** Comparison of four monitored lines of 25 m illustrating the increase in carrageenan between the start ($t_0$) and the end of a cultivation cycle ($t_{40}$). Carrageenan is expressed in weight per unit area ($g \cdot m^{-2}$). The area is defined by a neighborhood of a 20 cm radius around each pixel. The values at the bottom of each line indicate the total weight of carrageenan produced by the corresponding line.

## 4. Discussion

### 4.1. Drone-Based Detection of Kappaphycus at the Scale of Cultivation Lines

Drone multispectral imagery provided a valuable dataset for detecting and analyzing *Kappaphycus* biomass and biochemistry on floating long lines. These variables were estimated at the scale of single cultivation lines. The spatial distribution maps produced in

this study also enabled quantifying changes in *Kappaphycus* fresh weight and carrageenan weight, at regular time intervals within each cultivation cycle. Upscaling of in situ measurements of fresh weight and carrageenan weight allowed for an overview of the spatial distribution of *Kappaphycus* fresh weight and carrageenan weight within a cultivation plot. The fusion of drone and in situ data via empirical models was used for extracting information at a scale that cannot be achieved with satellite images. Nevertheless, the values of remotely sensed *Kappaphycus* biomass metrics obtained in this study were coherent with measurements reported in similar studies based on satellite imagery [17,18]. These previous studies used high spatial resolution satellite images of 1.8 m pixel size from GeoEye-1 [17] or 4.8 m from PlanetLabs [18], but these resolutions did not permit to identify individual long lines within a cultivation plot. To our knowledge, this study is providing, for the first time, an estimation of biomass and yield at the scale of lines by exploiting the very high spatial resolution of drone data with a pixel size of 1.6 cm. The drone spectral resolution was also important for classification accuracy, with five spectral bands, three in the visible and two in the near-infrared. Despite their higher absorption with water depth compared to visible wavelengths, the Red-Edge band at 717 nm and the near-infrared band at 842 nm were highly important for identifying floating seaweed with the machine learning algorithm. Indeed, *Kappaphycus* long lines often float in the sub-surface at a depth varying in their weight and the anchoring system, but seldom below ca. 30 cm in the cultivation plots of this study. As with all seaweeds, *Kappaphycus* is characterized by the absorption of incident light by photosynthetic and accessory pigments in the visible and a strong reflection in the near-infrared (Nurdin, comm. pers.; [30]). This spectral feature is detectable in seaweed observed at the spectral resolution of the DJI Phantom 4 multispectral camera [43]. Using a similar MicaSense RedEdge multispectral sensor mounted on a drone, ref. [33] applied vegetation indices using near-infrared bands to map the red macroalgae *Pyropia yezoensis* aquaculture on semi-floating nets in China. For satellite data, ocean color indices such as the Floating Algae Index exploit the difference between the reflectance in the near-infrared and a baseline formed by red and shortwave infrared bands [29]. In our study, the RF algorithm was trained to detect seaweed using the five spectral bands of the drone multispectral sensor plus a Blue/Red-Edge ratio. This band ratio is recommended for increasing the contrast between floating seaweed from the background seafloor and, thus, improving classification. The capacity of the algorithm to detect the class of *Kappaphycus* was good, as estimated by producer accuracies ranging from 90.5 to 93.8%. The algorithm was trained with various environmental and illumination conditions covering the three cultivation cycles and different vertical positions of the long lines from the surface to the subsurface.

### 4.2. Estimation of Biomass, Carrageenan and Production

Comparing the results obtained in this study (fresh weight, SGR, production, carrageenan weight), either from in situ data or drone-derived, with previous works should be cautiously carried out. Data can be collected for different seasons with different environmental conditions, culture methods, methodologies for carrageenan extraction and *Kappaphycus* strains (also called cultivars) with different genetic characteristics [7]. In our work, the three cultivation cycles spanned the months of March to July, mainly corresponding to East Monsoon conditions [51] with lower rainfall [18]. Water temperature, salinity, turbidity, pH and nutrients are environmental variables that influence the growth of *Kappaphycus* [13,15]. However, most abiotic parameters showed a low variability within and between cultivation cycles, and no trends were observed during the three cycles spanning the months of March to July 2022 (Supplementary Table S1). Globally, the water quality corresponded to optimal conditions for the aquaculture of *Kappaphycus* in Indonesia [16]. This is supported by the growth rates estimated from individual thalli ranging from 5.4 to 6.9%·d$^{-1}$, which correspond to high growth rates [12,17,52–54]. Water turbidity had the highest variability and can be a limiting factor for *Kappaphycus* growth by reducing the downwelling light for photosynthesis. High turbidity levels have also been reported

to be related to ice-ice disease outbreaks [13]. The drone estimation of fresh weight at harvest per meter of a line is consistent with the number reported by [18] of 2.24 kg·m$^{-1}$ used by the Department of Maritime Affairs and Fisheries of South Sulawesi to estimate farmers' production. In this study, an average of 2.15 kg·m$^{-1}$ of fresh weight was estimated for cycle 2 but lower values were obtained for the two other cycles. The fresh biomass is subsequently dried by farmers, an important step that can affect the quality and quantity of the carrageenan yield [55]. The requirement for selling raw dried seaweed to private companies for carrageenan processing is set by the Indonesian government at a maximum moisture content of 35% [10,56]. In this study, the fresh samples had an average moisture content of ca. 90%. The estimation of dry weight (DW) with 35% of moisture content per kilometer of line ranged from 0.22 tons DW·km$^{-1}$ for cycle 3 to 0.33 tons DW·km$^{-1}$ for cycle 2. If we consider five cycles per year and use the cycle 3 value to provide conservative estimates for cycles 4 and 5, the annual production can be estimated at 1.33 tons DW·km$^{-1}$. This result can be compared with the values provided by [10] who synthesized the production of dried seaweed for different cultivation systems in various countries. For Indonesia, the production per unit of cultivation line ranged from 0.55 to 1.68 tons DW·km$^{-1}$ per year for floating systems (opposed to the off-bottom cultivation technique). Drone-derived estimations for fresh weight and DW production per unit of cultivation line are therefore consistent with published values. However, the drone-derived growth rates (SGR) were slightly lower compared to the SGR estimated from individual thalli. This could be due to the limited sampling of individual thalli (*n* = 15) or individual lines by the drone (*n* = 4 to 8), but an underestimation of the fresh weight per line due to the misclassification of *Kappaphycus* by the algorithm cannot be excluded. In fact, the best in situ measurement to validate drone fresh weight estimations per line would be for future experiments to weigh lines and not individual thalli, as carried out by [7]. These authors reported production in different locations in Indonesia as the difference in fresh weight between the start and the end of cultivation cycles expressed in g·m$^{-1}$·cycle$^{-1}$. In this study, the drone estimates of production for cycles 1 and 2, respectively 1516 and 1807 g·m$^{-1}$, are comparable with their reported values for a similar period for the sites of Pangkep and Bantaeng in South Sulawesi [7]. The carrageenan content measured on samples collected during each cultivation cycle was consistent with values reported by [12] for Takalar Regency and corresponded to high levels. However, these values are not comparable with other results obtained with different drying techniques such as traditional sun-drying for 2–3 days and/or different extraction methods [57]. In this study, the samples were dried in the laboratory for 3 days at 60 °C, which explains why the range of values for carrageenan was higher than reported in studies using the traditional drying method used by the farmers [7,17]. The drone-derived carrageenan weight at harvest estimated at the scale of lines varied from 120 g·m$^{-1}$ for cycle 2 to 88 g·m$^{-1}$ for cycle 3. Considering five cycles per year and using the cycle 3 value to provide conservative estimates for cycles 4 and 5, the annual carrageenan production per kilometer of cultivation line could be estimated at 0.5 t·km$^{-1}$. When upscaled to a cultivation plot, an estimation of the carrageenan weight per unit area could be obtained (Figure 6D). For cultivation cycles 1 and 2, the carrageenan production was of ca. 4 t·ha$^{-1}$·cycle$^{-1}$, slightly more than twice the values reported by [7]. This difference could be due to the specific cultivation method used by the farmer at our study site in Takalar and captured by the drone (spacing of cultivars of 25 cm, spacing between lines of 50 cm).

### 4.3. Sources of Error and Constraints

Classification accuracy is one factor that is expected to contribute to the uncertainty of *Kappaphycus* fresh weight and carrageenan weight estimations from remotely sensed imagery. Although classification errors were minimal in this study, misclassification of pixels identified as *Kappaphycus* may lead to under-estimation of fresh weight, while misclassification of other classes as *Kappaphycus* would produce the opposite result. Therefore, it was important to consider predictor variables that maximized the separability between

the different classes. In this study, the Blue/Red-edge band ratio was introduced in order to enhance the differentiation between *Kappaphycus* and the background bottom. Selecting an effective training dataset covering the various illumination conditions was also important for increasing classification accuracy. Considering the average misclassification error of the *Kappaphycus* class, (approximately 10%) it is expected that drone-derived fresh weight estimations may contain a similar percentage of underestimation. The empirical relationship between seaweed area and FW showed a high and significant coefficient of determination ($R^2 = 0.8$), but with some variability estimated by the RMSE of 44 g and the MAE of 34 g. Assuming a final harvest weight of around 300 g, a variability of ca. 10% can be propagated to the output values. However, this variability can be reduced by improving the method to estimate the seaweed surface. Instead of using the length and width of the thallus, the surface should now be digitized from RGB photographs. In this study, the drone images of five flights could not be exploited due to sun glint and/or poor water surface conditions with waves that hindered the production of orthomosaics. In order to minimize sun glint, drone imagery should be acquired early in the morning or late in the afternoon when the sun is at an angle of less than 30° from the horizon. Unfavorable conditions of winds and waves should be avoided as they may cause changes in the position of the cultivation lines during the flights, therefore impacting the photogrammetric process. Turbidity strongly influences water's optical properties and may alter the detection of *Kappaphycus*. This depends on the position of seaweed in the water column ranging from almost no effect when they float at the surface to light being scattered by suspended particles when the lines are ca. 30 cm below the surface. Images should ideally not be acquired during turbid conditions, with an empirical recommendation to fly ideally when turbidity level is <20 NTU. Another important factor influencing the detection of *Kappaphycus* on drone images was the varying immersion depth of the cultivation lines due to increasing biomass, combined with the variable efficiency of the anchoring system, and changes due to tidal variation [25]. In fact, we could not use traditional vegetation indices such as the NDVI [29], in spite of the strong reflectance of *Kappaphycus* in the NIR. In this study, NDVI was significantly affected by variations in seaweed immersion depth and could not be used to determine the coverage of *Kappaphycus* per pixel nor to use it as a proxy for fresh weight. NDVI was also reported to be sensitive to the viewing geometry [29]. It is well known that NIR wavelengths are strongly attenuated due to the high pure seawater absorption coefficient. This is why a machine learning approach was chosen and trained with different immersion and background conditions. Machine learning based on a convolution neural network was similarly used to detect floating *Sargassum* [58]. In order to account for changes in immersion depth a first step could have been to take spectral measurements of water-leaving reflectance for the different conditions. A radiometric correction could have been applied prior to image analysis in order to enhance the classification outputs. However, in the perspective of developing a more generic methodology, we are considering the possibility to apply a radiative transfer model approach [59,60].

*4.4. Implications for Kappaphycus Aquaculture Management*

Optimal management of *Kappaphycus* cultivation requires repetitive and accurate estimations of yield metrics [9,10,16,18]. This study highlights the effectiveness of drone-based imagery in *Kappaphycus* aquaculture monitoring by providing fine-scale, multi-temporal information on the spatial distribution of fresh weight and carrageenan weight. Drone-based remote sensing allowed the measuring of the *Kappaphycus* fresh weight and carrageenan weight both at the level of cultivation plots and at the level of individual lines. Analysis of drone imagery assisted in visualizing along-line variability of fresh weight and carrageenan weight in unprecedented detail. This capability is considered very important for aquaculture managers as it enables them to monitor the growth of *Kappaphycus* at a fine scale and thus increasing the production efficiency and reducing the costs and potential environmental impacts on the crops [16]. Carrageenan weight estimation is one of the most important parameters in *Kappaphycus* aquaculture and drone mapping provides an

efficient and reliable tool for aquaculture-scale carrageenan estimation by exploiting a small number of in situ measurements. Drone-derived metrics about fresh weight and carrageenan weight at the long lines scale, are useful for upscaling overall production at different spatio-temporal scales. Here, total production and productivity rates were upscaled from a sample of long lines mapped with the drone to a small family-scale farm and a large-scale leader farm [10]. According to [57], a family-scale farm in South Sulawesi may own typically 6 km of long lines covering an area of ca. 0.56 ha while a large-scale farm consists of 30 km of long lines covering an area of 2.8 ha. In this study, it was found that with five cultivation cycles throughout a year, a family-scale farm would produce $8 \text{ t·yr}^{-1}$ of dry weight biomass (dried with 35% of moisture content) with a productivity of $1.3 \text{ t·km}^{-1}$ or $14.3 \text{ t·ha}^{-1}$. A large-scale farm would produce $39.9 \text{ t·yr}^{-1}$ with the same productivity rates. These numbers are in agreement with the estimations reported by [57]. However, when total production and productivity rates were upscaled from drone data estimated from a sample of three cultivation plots (i.e., extracting all biomass from a cultivation plot: Figure 6), the values were four times higher, with annual productivity of $5.1 \text{ t·km}^{-1}$ or $55.5 \text{ t·ha}^{-1}$ and production of 31.0 and $155.5 \text{ t·yr}^{-1}$ for, respectively, a small and a large farm. This difference is likely due to the specific cultivation method at our study site (spacing of cultivars and lines) observed by the drone. It was beyond the objective of this work to do an analysis of all cultivation plots from Takalar Regency but very high-resolution drone data provide an effective tool for a fine-scale analysis of farming areas. Drone mapping could be complementary to satellite remote sensing in helping the implementation of aquaculture policies [18]. Drone surveys provide imagery at centimeter spatial resolution, unaffected by cloud cover that can take place on-demand with an affordable off-the-shelf drone. Drones may help standardize data collection and analysis across farms and regions, improving the quality and consistency of the production data. Furthermore, this could be interesting for private companies processing carrageenan [9,10].

## 5. Conclusions

This study demonstrated the capability of drone imagery consisting of five multispectral bands to analyze the fresh weight and carrageenan weight of *Kappaphycus* during three consecutive cultivation cycles at a very high spatial resolution. Supervised classification using a random forest model was used to detect *Kappaphycus* on orthomosaics and discriminate it from background bottom and plastic buoys with high overall accuracy (>85%) across all temporal datasets. In situ measurements were necessary to calibrate classified pixels to fresh weight and carrageenan weight at the scale of cultivation lines. Growth metrics obtained from a sample of four to eight individual long lines agreed with previous studies. The drone estimate of the fresh weight of *Kappaphycus* at harvest varied from 1435 to $2151 \text{ g·m}^{-1}$, which is consistent with the value of $2240 \text{ g·m}^{-1}$ used by the authorities [18]. In addition, the net production of *Kappaphycus* was estimated at 1516 and $1807 \text{ g·m}^{-1}$ during two cultivation cycles, these values being representative of the average production in the wider region (South Sulawesi, [7]). An important perspective of drone remote sensing for *Kappaphycus* aquaculture is capturing potential temporal variations in seaweed biomass offering farmers a useful tool as an early warning of crop and yield conditions. By promptly addressing these issues, farmers can make data-driven decisions about when to harvest, or adjust cultivation practices and thus reduce the economic and environmental impact of their operations. Flexible and high spatio-temporal drone acquisitions open the way for *Kappaphycus* precision aquaculture.

**Supplementary Materials:** The following supporting information can be downloaded at: https://www.mdpi.com/article/10.3390/rs15143674/s1, Supplementary Table S1. Dates of the three cultivation cycles of *Kappaphycus* monitored in 2022 in Punaga (South Sulawesi) and corresponding water quality variables. Temp = Temperature, Sal = Salinity, NTU = Nephelometric Turbidity Unit, $NO_3^-$ = Nitrates, $PO_4^{3-}$ = Phosphates. Standard deviation in brackets. For carrageenan content, $n = 5$. NA = not available. Seawater variables were obtained for each cycle and sampling date. Water temperature was measured in situ using a digital thermometer, along with salinity measurements carried out using a digital refractometer. Water samples were stored in 1 L bottles and kept in a cool box during transportation from the field to the laboratory for pH, turbidity, nitrate ($NO_3^-$) and phosphate ($PO_4^{3-}$) analysis. Turbidity was measured with a turbidimeter and expressed in Nephelometric Turbidity Units (NTU). Nutrient concentrations (mg·L$^{-1}$) were measured using a DREL 2800 spectrophotometer and processed with the methods described by [61].

**Author Contributions:** Conceptualization, N.N. and L.B.; Methodology, N.N., T.K. and L.B.; Software, S.O.; Validation, E.A.; Formal analysis, E.A. and G.B.; Resources, R.S., H.A., E.N.Z. and G.B.; Data curation, A.A.; Writing—original draft, E.A.; Writing—review & editing, E.A.; Visualization, A.A.; Supervision, N.N. and L.B. All authors have read and agreed to the published version of the manuscript.

**Funding:** This work was partially supported by the "PHC NUSANTARA" program (project number: 47060PG), funded by the French Ministry for Europe and Foreign Affairs, the French Ministry for Higher Education and Research and the Ministry of Education and Culture, Riset and Technology of Indonesia (partner).

**Data Availability Statement:** Data are available on request from the authors.

**Acknowledgments:** We thank Ibu Fritriani from the water quality laboratory of the Faculty of Marine Sciences and Fisheries of Hasanuddin University for the analysis of carrageenan content and all the information she provided about the analysis. We would also like to thank Bede F.R. Davies for proof-reading and providing feedback on the manuscript.

**Conflicts of Interest:** The authors declare no conflict of interest.

## Appendix A

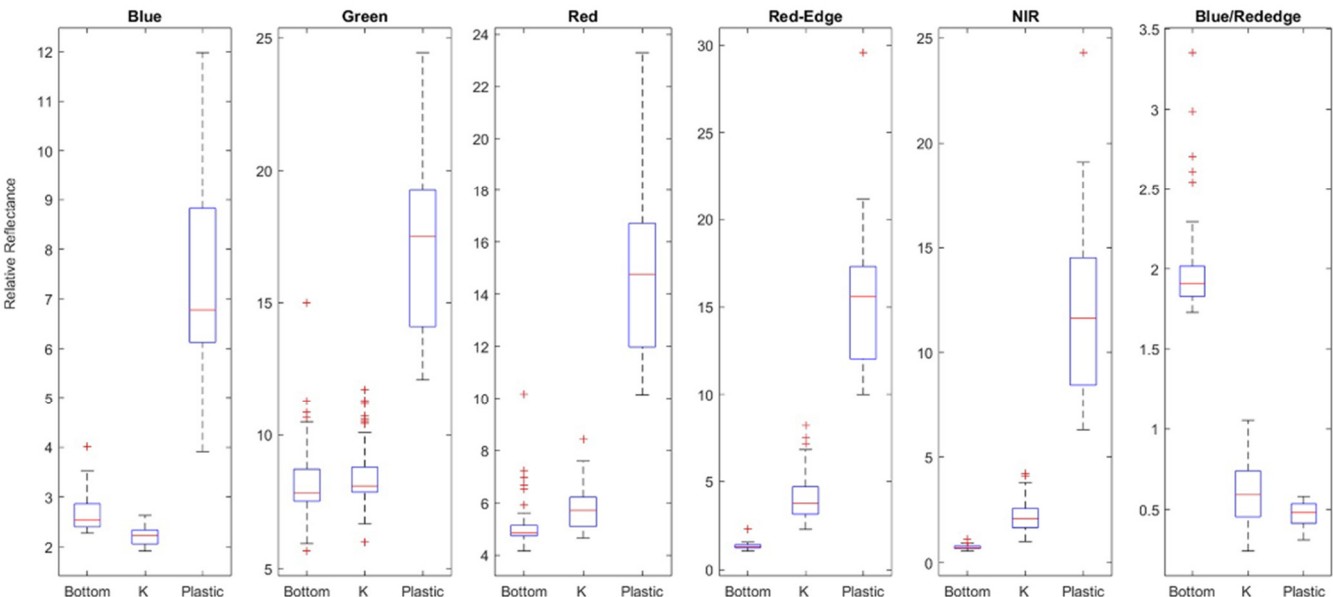

**Figure A1.** Boxplots resulted from an example training set showing the class separation at each of the six predictor orthomosaics (K = *Kappaphycus*). Crosses indicate outliers. The bottom and top of the blue rectangles represent the 25th and 75th percentiles respectively, whereas the red line indicates the median value. The whiskers extend to the minimum and maximum values that are not considered outliers (i.e., they are no more than $\pm 2.7\ \sigma$ apart).

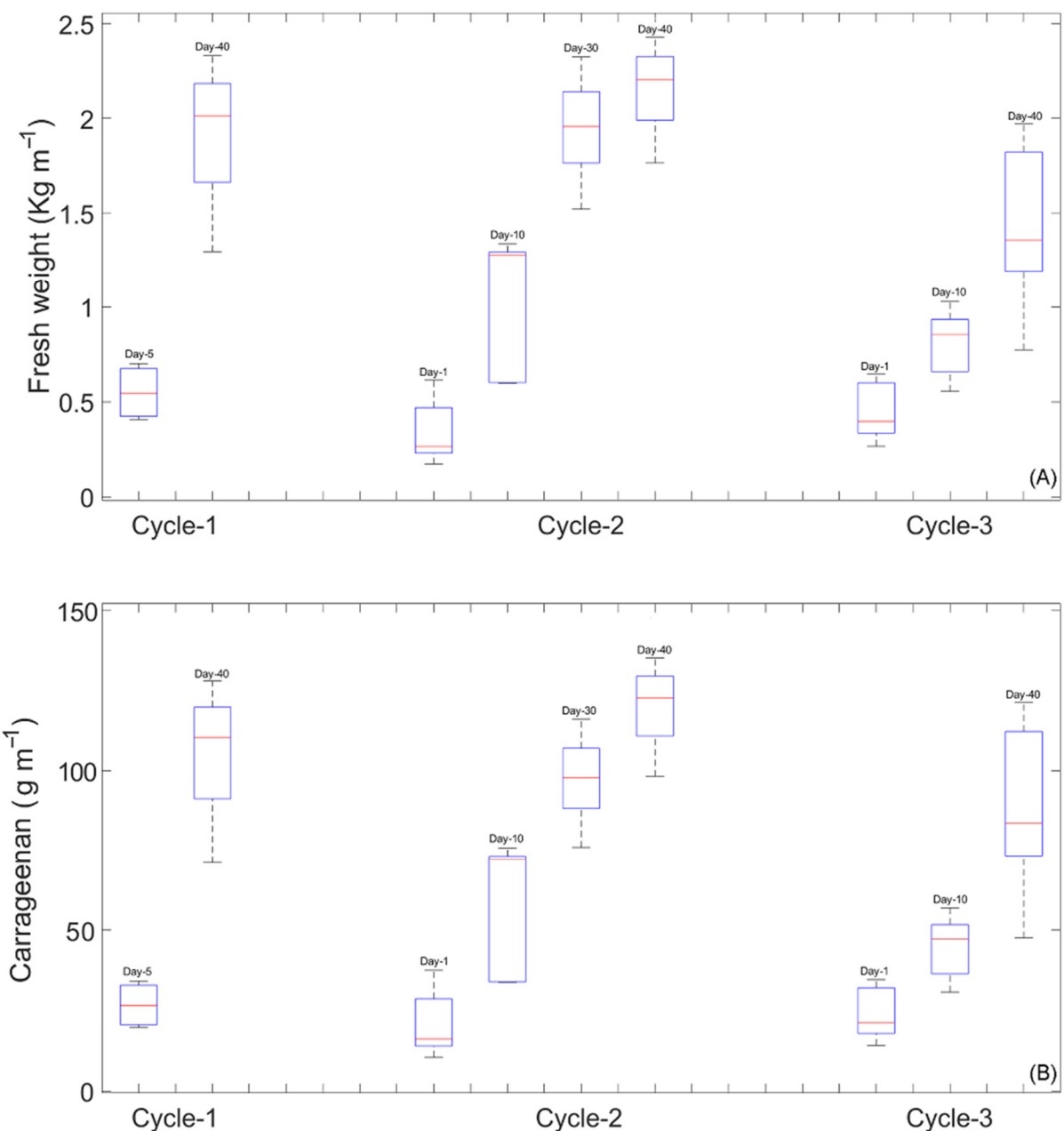

**Figure A2.** *Kappaphycus* increase in fresh weight (**A**) and carrageenan weight (**B**) during three cultivation cycles in Punaga (South Sulawesi) in 2022 estimated from drone imagery. The weights are expressed per linear meter of cultivation lines. Boxplots resulted from samples of 4–8 individual monitoring lines. The bottom and top of the blue rectangles represent the 25th and 75th percentiles respectively, whereas the red line indicates the median value. The whiskers extend to the minimum and maximum values that are not considered outliers (i.e., they are no more than ±2.7 σ apart).

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
