# Peer review of "Precision Aquaculture Drone Mapping of the Spatial Distribution of Kappaphycus alvarezii Biomass and Carrageenan"

_remotesensing, doi:10.3390/rs15143674_

Round 1

Reviewer 1 Report

Please see my comments in the attached file. 

I have no comments on English language quality at this round.

Author Response

We would like to thank the reviewer for the supportive comments and constructive feedback.

Please find below our responses (in bold) for each one of the comments.

 Major points:

- Line number must be added to the manuscript to enable a faster and easier tracking of the contents.

Line numbers added.

- The author needs to rethink and restructure the contents of the manuscript. Does this study focus on drone based mapping and estimation techniques for cultivated seaweed at farm scales or including seaweed cultivation (water quality variation, growth rate…)? Since you are submitting the manuscript to Remote Sensing journal, I suggest restructuring the contents and only concentrating on drone image processing techniques for cultivated seaweed mapping, fresh biomass and carrageenan estimation, and raising the novel contribution of the manuscript to remote sensing community.

Thanks for pointing out this issue. We have now restructured several parts in the Introduction section in order to better address the remote sensing character of this study. Lines 62-69; 74-75; 79-84; 110-114.

- Paragraph “One major challenge…. to high-resolution multispectral data”: I don't see the actual link between the current challenges in seaweed cultivation and the need of using either remote sensing or drone. Please provide additional arguments to clarify this point. Also, the description of water quality management and cultivation techniques should be reduced in words and focus on the linkages between cultivation characteristics and drone based detection/ mapping.

Water quality information is now minimized and reduced to a single supplementary table (Lines 659-669).

We have now focused more on the challenges which can be addressed by remote sensing. Lines 79-84.

- The author does not use the drone for water quality/ water management assessment. I don’t think the section of water quality description fits the aims and contents of this manuscript.

The reviewer made a point: we agree that the manuscript would benefit from restructuring by focusing on the remote sensing parts and specifically giving less importance to the water quality aspects. We, therefore, removed paragraph 2.6. “Environmental parameters” from the Material & Methods section, also removed the first paragraph of the results 3.1.”Environmental data”, and lastly all the columns related to water quality from Table 1. We moved all this to a single table in the supplementary material (Lines 659-669) as this may be of interest to a broader readership, and kept a line in the discussion referring to it (Lines 476-478).

- Introduction section, please clarify the actual needs of using drone image for this case study (i.e., seaweed cultivation). This research was conducted as due to less/ no one having done it previously? You mentioned the reason only in this statement “The application of drones in aquaculture appears to have a strong potential due to their flexibility in data acquisition and the high resolution of resulting imagery.” That is not sufficient.

We have now included the main reasons for drone use in this study. Lines 79-84.

- Introduction section, paragraph of “remote sensing” should focus more on drone characteristics, which help to detect the seaweed farming at different scales, and cultivation features.

We added more information about the characteristics of drone sensors regarding seaweed mapping. Lines 95-97.

- Introduction section, what are the disadvantages/ drawbacks of previous studies using drones for submerged vegetation/ seaweed mapping and assessment? What is the novel contribution of this study?

This study did not aim at improving drawbacks in previous studies. However, the novelty of this study is that it presents seaweed aquaculture monitoring as a novel application of drone remote sensing. Our study highlights the benefits of applying standard technology (drone and MS sensor) on a new mapping field.

- Figure 5: Model seems to have a level of uncertainty for the ranges 200 - 500 cm2. Do you support an estimation of these uncertainties?

We agree. It is true that we did not propagate this uncertainty to the final results.

We provide RMSE and MAE to estimate the average uncertainty of the model.

- The author only estimates the cover area of seaweed, builds an empirical model with uncertainty for a wide range of area (200 - 500 cm2, the data points are pretty far from the central line (red color)) to estimate the fresh biomass and other biometrics. This approach is convenient, however it is not consistent enough to use this approach for the retrieval of SGR (Table 3) and carrageenan content. How do you deal with the uncertainty of the estimation model in this study?

We recognize the variability in the empirical model, but we trust that the model is sound with a coefficient of determination of R² = 0.81. We calculated the RMSE and MAE to estimate the average uncertainty. It is true that we did not propagate this uncertainty. We added some lines in the discussion to acknowledge that and use the RMSE/MAE to give an idea of the error to the reader. In fact, it is suggested that this variability can be significantly reduced by digitizing the seaweed surface instead of using the length and the width. Lines: 543-549:

- Section 2.4: - Why does the author use the ML Random Forest (RF) model for the classification in this manuscript? There are better ML models out there for the classification works, such as Extreme Gradient Boost or CatBoost models. Any reason for using RF should be carefully clarified in this section.

The main reasons for using the RF are mentioned in Lines 237-238, and these include the speed of the algorithm, its availability across GIS software, and the insensitivity to overfitting.

- How do you split data into the training/testing dataset? What is the ratio of train/test dataset? Please clarify.

Lines 260-268: A few hundred pixels were extracted from the boundaries of the classified objects by applying a quasi-regular type of sampling (Fig. 3B). Selecting polygons with several pixels assists in having a robust training set; however, selecting polygons for validation would lead to underestimation of misclassifications in this study. Therefore, by selecting single pixels, the objectivity of the validation procedure was increased and spurious results could be captured effectively.

The percentage of training/validation pixels is 75/25 approximately (Line 268).

- Do you think 1000 decision trees too much for the RF implementation for seaweed from a 1,3 cm image? Use of high to very high decision trees in the RF model might lead to an overestimation in the prediction phase.

A value of 1000 trees is required for ensuring robust training of the model when several predictive variables are used (6 grids in this study). It is well-studied that the RF error is minimized when using many trees without the risk of overfitting. It is generally suggested that a larger number of trees produces more stable models and covariate importance estimates. Please consult the random forests literature for an extensive discussion of this parameter [e.g Strobl et al., 2009]

- This section should be splitted into sub-section of (i) model introduction; (ii) training/ validation data preparation; (iii) model implementation; (iv) model evaluation with formulas.

This study does not focus on the algorithm, and thus we consider that further splitting this section in subsections is not necessary. Splitting would also make the section look more fragmented which could be confusing for the reader.

- Discussion section 4.4, could you clarify a workflow in which the drone based monitoring can be applied for different farms and how complex the workflow is when applying at large scale. I think that point is a drawback of using drones in monitoring seaweed yield and cultivation.

In the Discussion, we mention that the presented approach can be applied across different farms using empirical data modeling, and does not depend on the size of the study area. This approach is not considered a drawback when applied at a large scale but rather assists in harmonizing the seaweed measurements in different aquaculture areas. Lines: 588-590; 610-612.

- Please have a comparison of derived carrageenan content in this study with in situ and other estimation in studies.

We compare in situ carrageenan content from previous studies in Lines: 511-514.

Minor points:

- Section 2.1, the author needs to tell the reader a bit more about the growth of seaweed and at which stages under farming conditions, the drone can detect and measure the biomass, carrageenan?

We added a sentence about when seaweed can be observed better with drone sensor. Lines 168-170.

- “Fresh seaweed was cleaned and washed with freshwater, cut into small pieces, and then soaked in water for 24 hours to remove salt and sediment. It was then dried as indicated previously. The analysis used sodium hydroxide extraction (NaOH) followed by precipitation with ethanol. More details can be found in [35].” The author needs, at least, to describe the way to have the carrageenan content in gram.

As suggested by the referee, we added more information on the protocol. Lines: 178-180.

Precipitated carrageenans were dried at 70°C for 4h. Dried carrageenan powder was weighed and compared to the dry weight of the seaweed.

- The author should provide additional details about the algorithm’s structure and advantages over other algorithms in Section 2.4.

We have now added more details about the functioning of the RF algorithm in Lines 222-231. The advantages are mentioned in Lines 233-234.

- ‘The RF implementation of ViGrA [44] was applied for class prediction in SAGA GIS:’ should be rewrite, something like “We used the ViGrA library to implement the RF in SAGA GIS.

The sentence has been rephrased. Line 250.

- Section 2.4: How does the author select the validation dataset? I do not see the percentage of training/ validation dataset and the techniques the author used to have the dataset.

Lines 259-265: A few hundred pixels were extracted from the boundaries of the classified objects by applying a quasi-regular type of sampling (Fig. 3B). Selecting polygons with several pixels assists in having a robust training set, because selecting polygons for validation would lead to underestimation of misclassifications in this study. Therefore, by selecting single pixels, the objectivity of the validation procedure was increased and spurious results could be captured effectively.

 The percentage of training/validation pixels is 75/25 approximately (Line 258).

- ‘Carrageenan was measured according to the Indonesian National Standard (SNI 01-4498-1998).’ Is it carrageenan in percentage or gram?

Carrageenan is expressed in percentage. This was already mentioned a few lines below the reference to the Indonesian National Standard in the first draft of the ms, but we added this information just after now. Lines: 174-175.

Reviewer 2 Report

This study explores the possibilities of using drones to monitor Kappaphycus alvarezii aquaculture, looking not just at the distribution and disposition of thali in the long-lines, but also to the changes in biomass and carrageenan content.

I like the paper very much, I think it is well written, the initial hypothesis are well stated, the material and methods described in detail, and I also think the results are discussed properly, although in my opinion some details on the extrapolation to cultivation productivity at national scale, or even between small or big scale farms, could be omitted.

I also wondered, if you have tried to check for relationships between SGR or Carragenean content with the environmental variables, with a multiple regression for example. This might bring new interesting results, although I understand that this might be out of the initial scopus of this paper, since you measure the environmental conditions, it would be nice to see how those relate (or not) to SGR and Carragenean content.

I only have some minor details about English wording, mainly in the introduction, which are detailed bellow:

Introduction

“Indonesia is the second largest seaweed producer in the world after China, but is the largest one of red seaweed” Rephrase, for example “…. and the first one when talking about red seaweed”

“Carefully monitoring and management of environmental factors…” How can you manage the environmental factors? I guess that what you can control is the management of the cultivation, Maybe you can rephrase it for something like “A careful monitoring of environmental factors which allow to detect stressful conditions and generate management adaptations of the cultivation are therefore required to better control biomass production and carrageenan content”

“There is a striking contrast with agriculture, where drones have been increasingly used to monitor crops, gather data and make better decisions about managing the fields.” Move this sentence to the next paragraph, after you introduce remote sensing techniques.

Author Response

We would like to thank the reviewer for the supportive comments and constructive feedback.

Please find below our responses for each one of the comments.

Comments for the authors

‘’ ..also wondered, if you have tried to check for relationships between SGR or Carragenean content with the environmental variables, with a multiple regression for example. This might bring new interesting results, although I understand that this might be out of the initial scopus of this paper, since you measure the environmental conditions, it would be nice to see how those relate (or not) to SGR and Carragenean content.’’

We did not apply any further analysis of the link between SGR and carrageenan content, nor with environmental variables. We are aware that we may have a correlation between SGR and carrageenan, but we did not want to explore this further regarding the remarks of referee 1 who pointed out that this would have been out of the scope of the journal. In fact, as referee one suggested we even removed all the water quality results from the main text to put it as a supplementary material. We restructured the manuscript by focusing more on the remote sensing parts and specifically giving less importance to the water quality aspects which include a small number of data that do not show any particular trend. The suggested approach would be something for a future study where more environmental data are available.

We think that the extrapolation to small and large farms is important to connect our work with previous publications and FAO reports.

Comments on the Quality of English Language

All suggested corrections were applied accordingly.

  • Lines: 46-47.
  • Lines: 74-76.
  • Lines: 110-114.

Reviewer 3 Report

Please check the attached comments

Author Response

We would like to thank the reviewer for the supportive comments and constructive feedback.

Please find below our responses for each one of the comments.

 Specific comments:

  1. In the introduction, it may be necessary to add why this research is important. My suggestion is also to add a discussion about the cost and production value of Kappaphycus seaweed so that it can add reasons to continue research on this topic.

We have now added the main challenges that highlight the importance of this research (Lines: 79-84).

Regarding the value, thank you for the suggestion but it is already obvious in the text that the Kappaphycus is a valuable seaweed (Lines: 60-63). 

  1. What is the method of selecting the cultivation plots sampled in this study?

The cultivation plots were selected in agreement with the local farmer. Particularly we selected specific cultivation lines both from the boundaries and the central part of the plot in order to minimize placement bias and obtain more representative information about seaweed biomass and carrageenan.   

Round 2

Reviewer 1 Report

Please see my comments in the attached file.

Only a minor editing is required. 

Author Response

We would like to thank the reviewer for the new comments and constructive feedback.
